# Polychromy in the Iberian Sculptures of Cerrillo Blanco: Analytical Study, Historical Context and State of Conservation

Julio Romero-Noguera [1,*], María Belén Ruiz-Ruiz [2], María Teresa Doménech-Carbó [3]
and Fernando Bolívar-Galiano [2]

1   Department of Painting, University of Seville, Laraña 3, 41003 Seville, Spain
2   Department of Painting, University of Granada, Avda. Andalucía s/n, 18014 Granada, Spain;
    belenrr@ugr.es (M.B.R.-R.); fbolivar@ugr.es (F.B.-G.)
3   Instituto de Restauración del Patrimonio, Universitat Politècnica de València, Camino de Vera, s/n,
    46022 Valencia, Spain; tdomenec@crbc.upv.es
*   Correspondence: juliorn@us.es

**Abstract:** In the environs of the city of Ipolca, today's town of Porcuna (Jaén), the Iberian civilisation left behind one of the most outstanding sculptural ensembles of Antiquity, made up of 27 groups of figures and hundreds of fragments dating from the 7th to the 2nd centuries BC. Despite its great relevance, there are very few scientific studies that serve as a basis for understanding the many questions that remain about how they were made, their significance, and their relationship to the culture that gave rise to them. This article studies the polychrome techniques used in the sculptures and puts them into context in Iberian art. The research has been carried out on original pieces from the Archaeological Museum of Jaén using stereoscopic optical microscopy (SOM), petrographic microscopy (PM), Fourier-transform infrared spectroscopy (FTIR), and scanning electron microscopy-energy dispersive X-ray analysis (SEM-EDX).

**Keywords:** Cerrillo Blanco; sculptures; Iberian; materials; polychromy; analysis

## 1. Introduction

The archaeological complex of Cerrillo Blanco is one of the main pinnacles of Iberian art. It comes from the site of the same name, in today's municipal area of Porcuna (Jaén, Spain), and is made up of 27 sculptural groups made of white calcarenite stone [1] in homage to a dominant dynasty in Ipolca, the capital of the Turdili, around the first half of the 5th century BC [2,3]. The sculptures were produced for a local aristocracy and depict various fights between men, animals, mythological beings and ceremonial figures (Figure 1) in order to satisfy their desire to be represented and receive homage in the earthly and spiritual worlds [4–6]. They are currently on display at the Museum of Jaén. Despite the historical and artistic importance of these sculptural ensembles, the materials and techniques used to make them are not well-known.

### Polychromy in Iberian Sculpture

The studies carried out on the Iberian sculptures found to date indicate that, in their original state, they were usually polychrome [7]. Everything seems to indicate that the colour had a markedly symbolic meaning unrelated to naturalistic depictions and that its use must have been strictly regulated, associated with funeral rites and their relationship with the transcendental world [8].

Turning to the matter of technique and procedure, the polychrome sculpture of any period usually involves three layers: the support; the preparation layer—which isolates the paint from the base and adds a clear, even base; and the pictorial layer, composed of pigments and binder materials [9]. The most relevant studies on Iberian sculpture show that, in general, the procedures and pigments applied were the ones known in the

Mediterranean environment in Antiquity [10]. As a sculptural material, soft rocks were used almost exclusively, especially limestone and sandstone. The presence of a gypsum preparation layer has been documented in such relevant works as the *Dama de Baza* (*Lady of Baza*) and the *Dama de Elche* (*Lady of Elche*) [11,12], though in other cases, it has not been described and the polychromy seems to have been applied directly onto the stone [13].

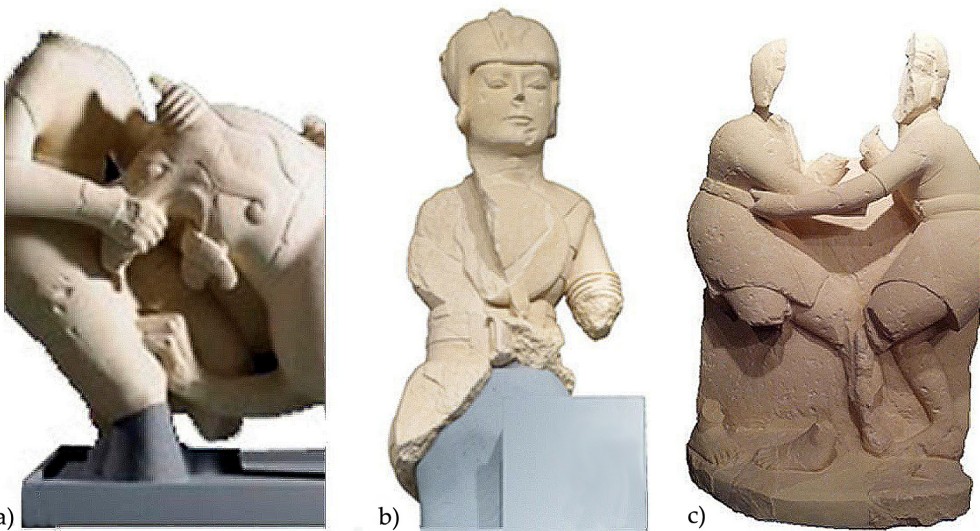

**Figure 1.** Sculptures from the Cerrillo Blanco complex. (**a**) Fight scene with griffin; (**b**) double-armoured warrior; (**c**) wrestlers, arena scene.

The pigments used were fundamentally the kinds available in the environment, especially earth tones based on iron oxides ($Fe_2O_3$) and clays (limonite and haematite, red ochre, yellow ochre and umbra), providing a wide variety of red, brown and yellow tones. Black pigments were also used, such as charcoal (carbon black) or burnt bones and ground white calcite. Red, green and blue colours were added to this basic palette, which were used in the Iberian world thanks to trade with the eastern Mediterranean civilisations [14]. Red vermilion or cinnabar (HgS) particularly stands out, which was well-known in the peninsula in Iberian times, with important deposits such as the mines of Sisapo, and Egyptian blue ($CaCu-Si_4O_{10}$), which is one of the oldest artificial pigments [15,16]. The use of malachite and azurite has also been documented [17]. As for the possible binders used, little is known to date.

In the case of the sculptures at Cerrillo Blanco, there is no information in this regard prior to the publication of this article. The Andalusian Institute of Historical Heritage (IAPH) carried out an analysis of their materials prior to their restoration [18], in which this matter was not addressed, despite the presence of traces of red colour on some pieces. This article is the first study of the possible techniques of polychromy used in the sculptures, their historical contextualisation and state of conservation.

## 2. Materials and Methods

The selection of samples was carried out after studying the sculptures and dozens of fragments deposited in the Museum of Jaén's storage. The presence of polychrome, which is always reddish in tone, can be seen with the naked eye on various pieces. Given the homogeneity of the materials and their state of conservation, the sculpture *Oferente con cápridos* (*Bearer offering caprids*) displayed in the museum was selected (Figures 2 and 3), as well as three fragments deposited in storage with numbers 0092, 0093 and 0097 according to the DOMUS (Digest of Museum Statistics) system for collections (Figure 3). Sample 0092 belongs to a large sculpture of a zoomorphic figure (part of the thigh and leg), and samples 0093 and 0097 are undetermined fragments. The sampling was carried out with a scalpel, following the UNE-EN 16085 standard [19], scraping the surface of the pieces and

obtaining microfragments to determine the type and composition of the possible strata that make up the polychromy.

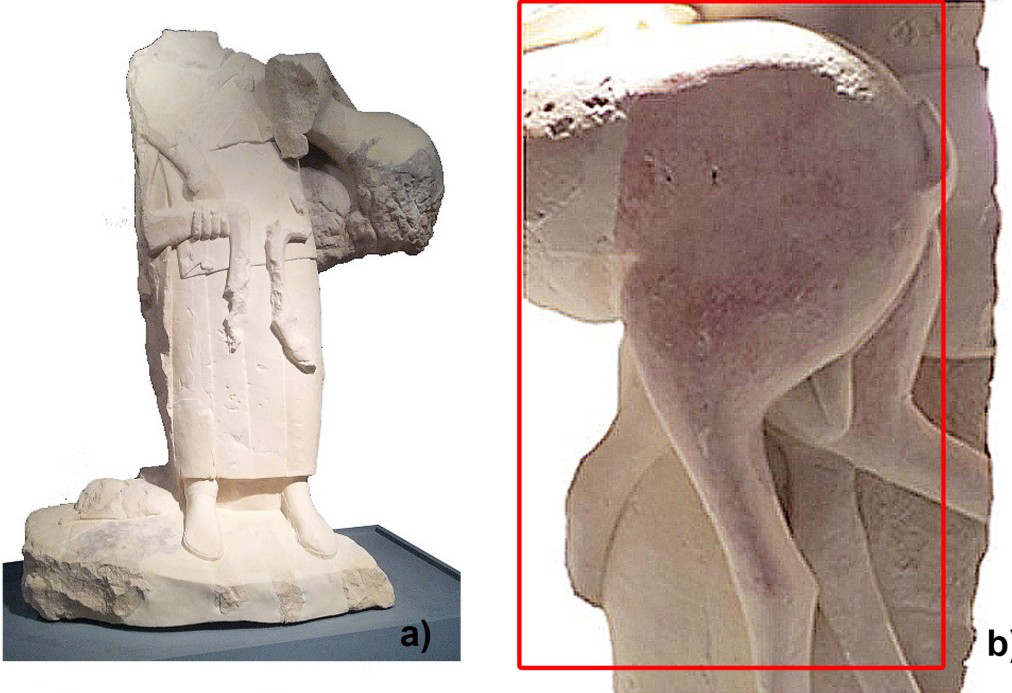

**Figure 2.** *Bearer offering caprids*. Obverse view (**a**) and detail of polychromy on the reverse side (**b**).

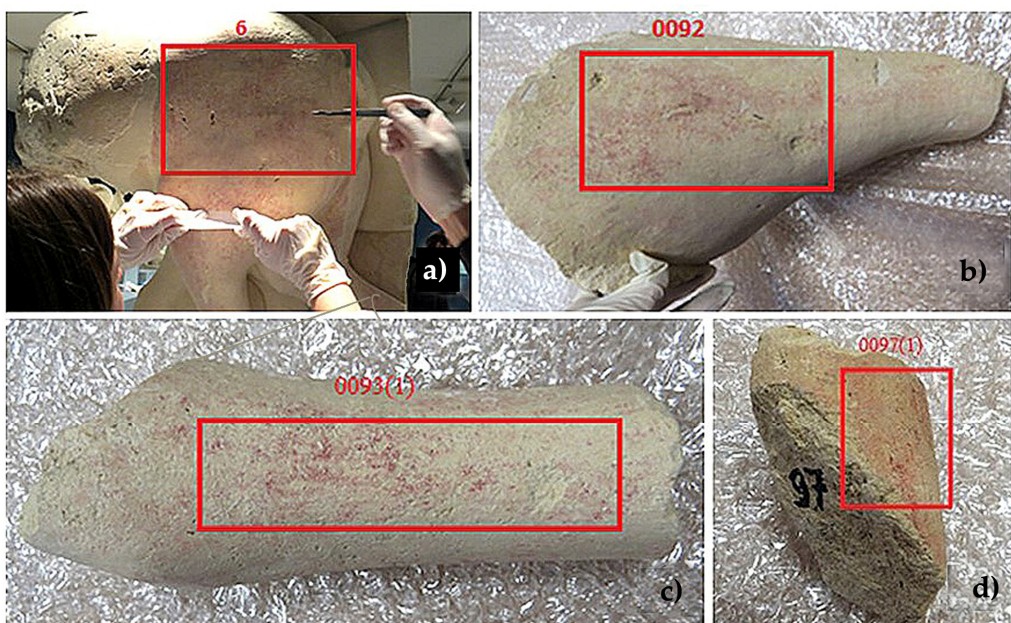

**Figure 3.** Samples obtained from pieces 6 (**a**), 92 (**b**), 93 (**c**), 97 (**d**) with traces of polychromy.

## 2.1. Instrumentation

The following instrumental techniques were used: stereoscopic optical microscopy; (SOM) and petrographic microscopy (PM); scanning electron microscopy with energy dispersive X-ray analysis (SEM-EDX), which provides information on the elemental composition and topographic images of the samples, and Fourier-transform infrared spectroscopy (FTIR), which enables the inorganic and organic components in the sample to be characterised.

### 2.1.1. Stereoscopic Optical Microscopy (SOM)

Samples of the pigments properly disaggregated and directly mounted on glass slides were examined with a Leica (Wetzlar, Germany) S8AP0 (X10-X80) that works with a Leica Digital FireWire Camera (DFC). Leica Application Suite 2.5.975 (LAS) was the software used.

### 2.1.2. Petrographic Microscopy (PM)

The samples were observed under a Carl Zeiss (Oberkochen. Germany) Jena Pol-U polarised light optical microscope equipped with a Nikon (Tokyo, Japan) D7000 digital microphotography unit, for mineralogical and textural identification. Images were acquired with cross-polarised light. To improve the quality of the image, the stage of the microscope was rotated until the maximum brightness of the red grains of pigment was reached. The spinning angle was measured using as reference a quartz grain of the mounted sample placed in the centre of the stage at the extinction angle.

### 2.1.3. Scanning Electron Microscopy-Energy Dispersive X-ray Analysis (SEM-EDX)

Microsamples excised from the polychromed parts of the sculptures were directly fixed on a metallic pin-stub holder with double-sided adhesive made of a graphite composite material.

The analysis was performed with a Jeol (Tokyo, Japan) JSM 6300 scanning electron microscope working with a Link-Oxford-Isis X-ray microanalysis system (Oxford Instruments, High Wycombe, UK). The analytical conditions were 20 kV accelerating voltage, $2 \times 10^{-9}$ A beam current and 15 mm as the working distance. Quantification was carried out in small areas of the samples using the ZAF method for correction of inter-element effects based on three factors: atomic number (Z), absorption (A) and fluorescence (F). In addition, spot measurements were made on individual grains and aggregates to obtain an estimate of their stoichiometric composition. All the X-ray microanalysis data were processed with the Inca 5.1 (Link-Oxford-Isis) software. In the heterogeneous stone support composed of several minerals, X-ray spectra were acquired in area mode. The acquisition area varied in order to always maintain it over the pattern of heterogeneities of the stone. This procedure is accompanied by spot analysis in individual grains of all minerals composing the outer and irregular pigment layer. Quantification was carried out using the ZAF method for the correction of inter-element effects based on three factors: atomic number (Z), absorption (A), and fluorescence (F). The standard deviation was calculated from three measurements performed in spot or area mode. This statistical value provided a picture of the degree of heterogeneity of the red pigment. More details on the working conditions used are summarised in Table S1 provided as Supplementary Electronic Material.

### 2.1.4. Fourier-Transform Infrared Spectroscopy (FTIR)

The analysis was carried out with a Bruker ALPHA portable spectroscope equipped with reflectance, diamond ATR and transmittance accessories. The software used was Opus 7.0 with the following operating parameters: 4 cm$^{-1}$ resolution, 256 scans, 7500–400 cm$^{-1}$ range and reflectance mode in a gold mirror. A potassium bromide (KBr) tablet was prepared and pressed at 10 tons for 10 s in a Perkin Elmer (Waltham, MA, USA) hydraulic press.

Samples of selected polychromed areas of the sculptures were powdered in an agate mortar and directly measured with an MKII Golden Gate attenuated total reflectance mode (ATR) accessory of a Vertex 70 (Bruker Optics, Billerica, MA, USA) FTIR spectrometer that operates with a detector with temperature stabilised using FRDGTS (fast recovery deuterated triglycine sulfate) Bruker Óptica®. The working conditions were as follows: number of scans, 32; resolution, 4 cm$^{-1}$. The data were processed with OPUS/IR software, version 5.0.

## 3. Results

### 3.1. Optical Microscopy

The images obtained at low magnification that are displayed in Figure 4 show the micromorphology of the red polychromed areas. The samples are irregular cryptocrystalline

aggregates of the calcite that forms the biocalcarenite rock used in the sculptures [1]. The dark red pigment appears as a thin layer of cryptocrystalline morphology deposited directly on the surface of the stony samples without any intermediate preparation or ground layer. In sample 0092, a bright, thin layer of synthetic organic consolidant can be seen that impregnates the surface of the sample (Figure 4b). Previous reports about the restoration of the Cerrillo Blanco sculptures indicate that the most eroded areas of many sculptures were consolidated with Paraloid B72 [1,18]. Consolidant was seen to be lacking on the back of the sample of a few hundred μm, suggesting low penetration achieved in the consolidation treatment and a low deterioration risk of the original material in the short term, due to the low quantity applied.

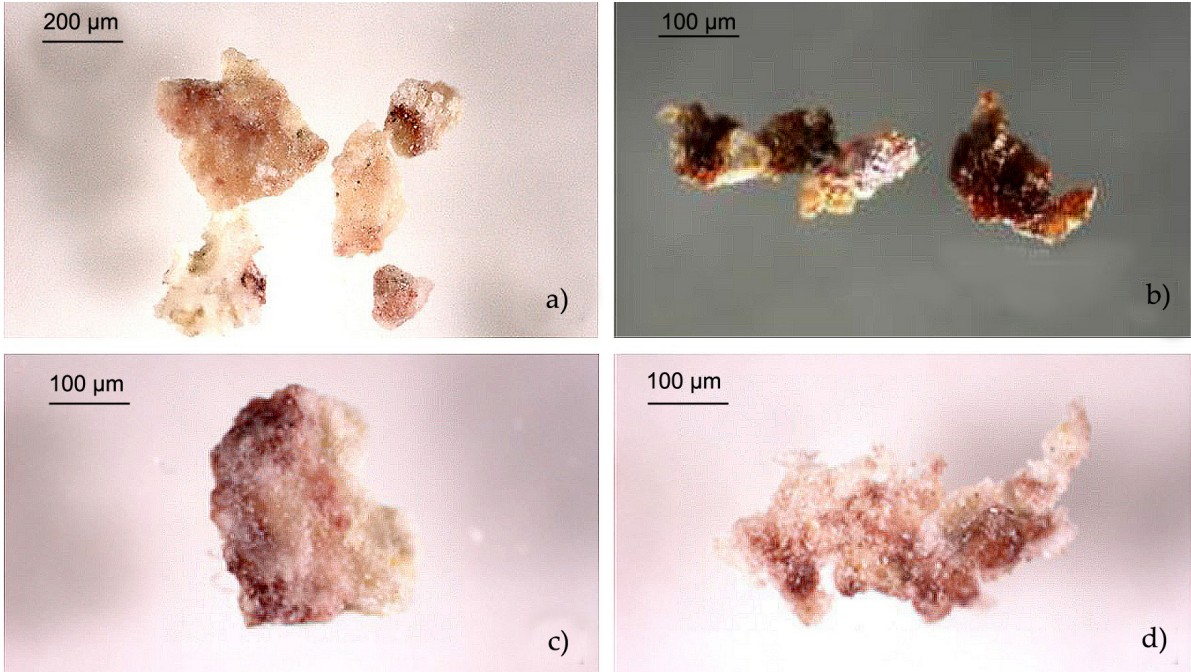

**Figure 4.** Images, obtained using reflected light of the polychromed layer of samples (**a**) 6 (50×), (**b**) 0092 (10×), (**c**) 0093, and (**d**) 0097 (50×).

Examination of the samples in transmitted polarised light confirms the previously described findings (Figure 5). The red pigment grains are below 100 μm in size, which is characteristic of clayey minerals. They form clusters or aggregates fixed to larger calcite or quartz grains, and they often exhibit a characteristic high relief [20].

Under cross-polarised light, the samples look like irregular stony fragments below 100 μm in size with the outer layer dyed in red. The pigment partially covers the surface of the sample and forms cryptocrystalline clusters or agglomerates. When the sample could be disaggregated and red agglomerates of red pigment could be isolated, they exhibited characteristic hues that range from deep orange to dark red, which indicates that the pigment is coloured primarily by haematite (Figure 5) [21,22]. The agglomerates also exhibited high relief typical of haematite (refractive index = 2.74 − 2.78) [22]. In some samples, yellow-orange agglomerates were also accompanying the red particles (Figure 5b). They have been associated with other iron oxide minerals. These mixtures have been reported as characteristic of red earths [22]. Translucent grains, also identified in the mounted samples, are ascribed to accessory minerals such as quartz and calcite. They could be part of the biocalcarenite support, although they are also commonly found in red earth pigments.

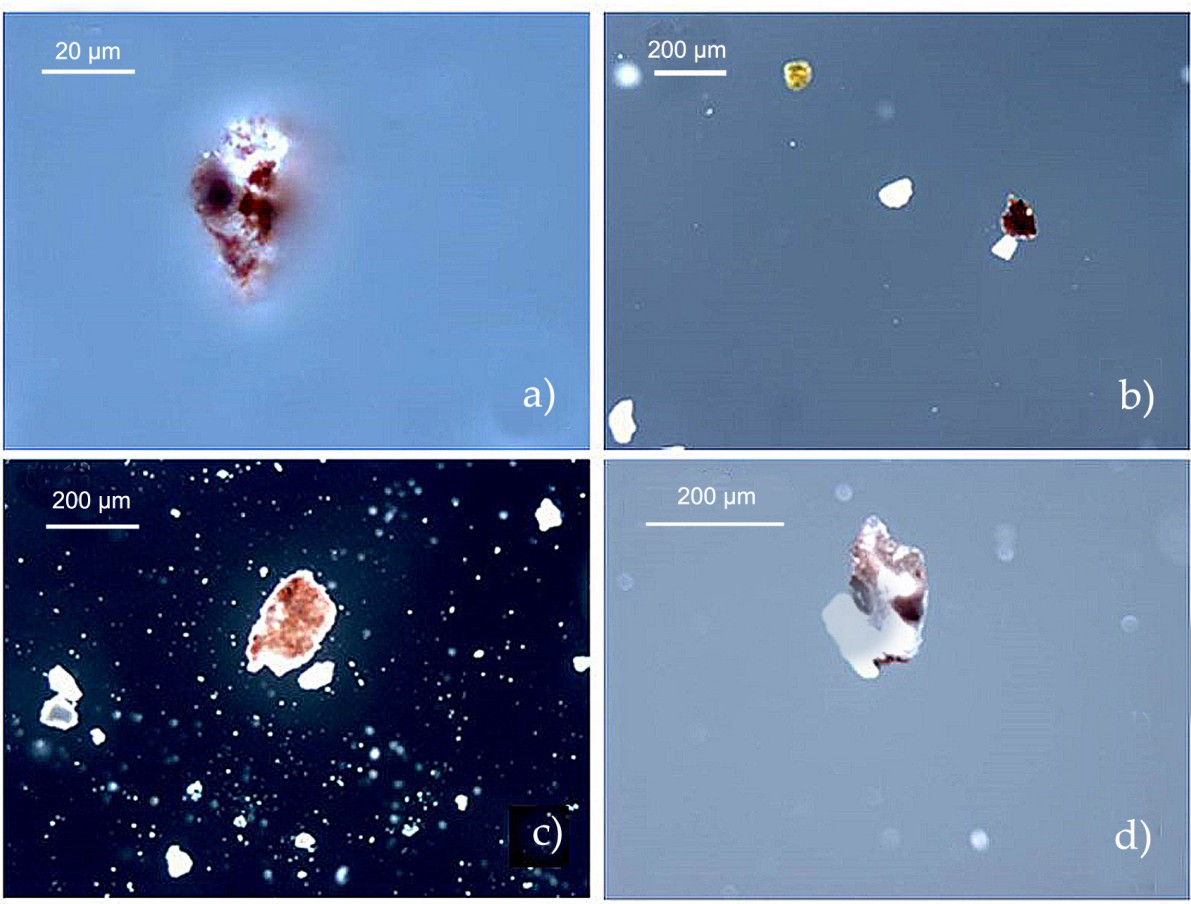

**Figure 5.** Image obtained using transmitted light of samples (**a**) 6 (75°), (**b**) 0092, and (**c**) 0093 using polarised light with Nicols partially crossed (90°), and (**d**) sample 0097 using polarised light with Nicols partially crossed (80°).

*3.2. SEM-EDX*

The secondary electron images of the samples acquired using SEM (Figures 6 and 7) show the characteristic cryptocrystalline morphology of the biocalcarenite rock used as the support of the sculpture. Eventually, small laminar grains (<1 μm) (which are pointed out with arrows in Figure 6a–d) associated with clay minerals of dioctahedral 1:1 (kaolin) group were observed (see FTIR results). Acicular microcrystals were also identified in some of the samples such as 0092. They are present in clusters and small aggregates. In both cases, the individual needles are randomly crossed (see Figure 7a). Angular crystals associated with quartz and siliceous minerals are also identified, with a diameter greater than 2 μm in samples 0092 and 6 (Figure 7b) and around 1–2 μm in sample 0093. In sample 0097, biological structures are identified. They are probably fragments of skeletons of planktonic unicellular marine microalgae of the coccolithophore type, with average sizes between 1 and 5 μm and a calcareous composition. The internal soft structure of the coccolithophore cell is protected by calcified structures, which are disposed as a rounded shield called the coccosphere [23]. The elements of the coccosphere have the form of striated disks with a hole in the centre (see Figure 7c,d). Coccolithophores have been considered the major planktonic group of microorganisms, and they are the main microorganisms responsible for pelagic $CaCO_3$ production [24]. These microfossils take part of the biocalcarenite stone of the sculptures.

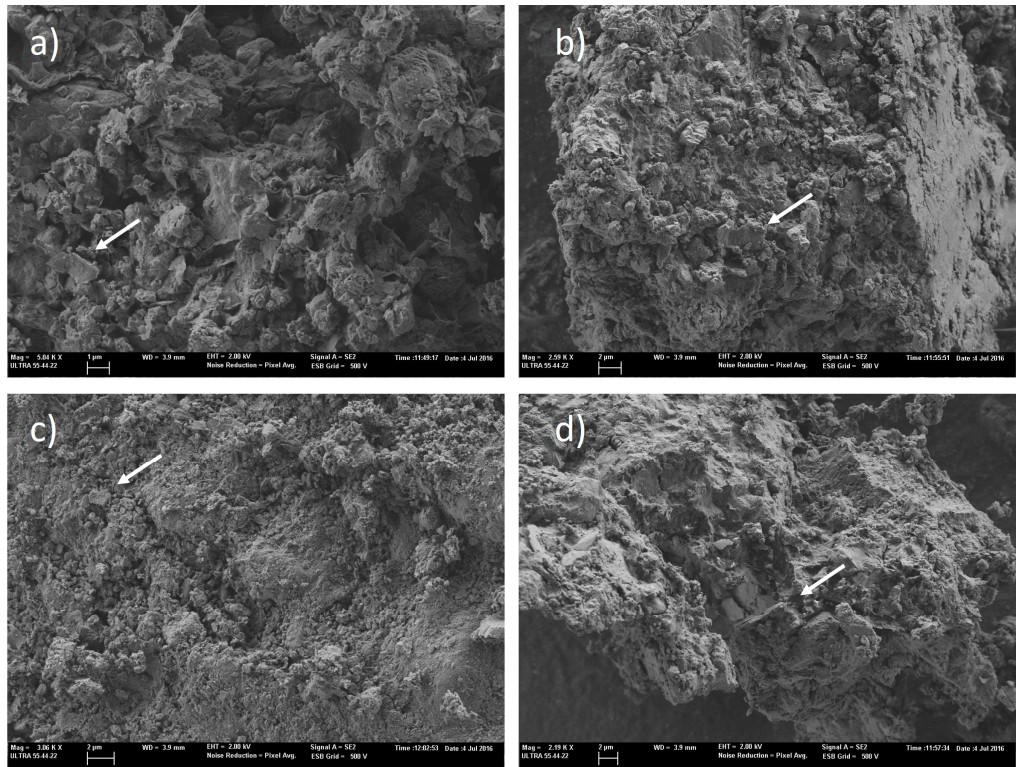

**Figure 6.** Secondary electron images acquired at 3 kV: (**a**) sample 6; (**b**) sample 0092; (**c**) sample 0093 and (**d**) sample 0097. The arrows in the figures point out grains with a laminar shape ascribed to dioctahedral 1:1 clay.

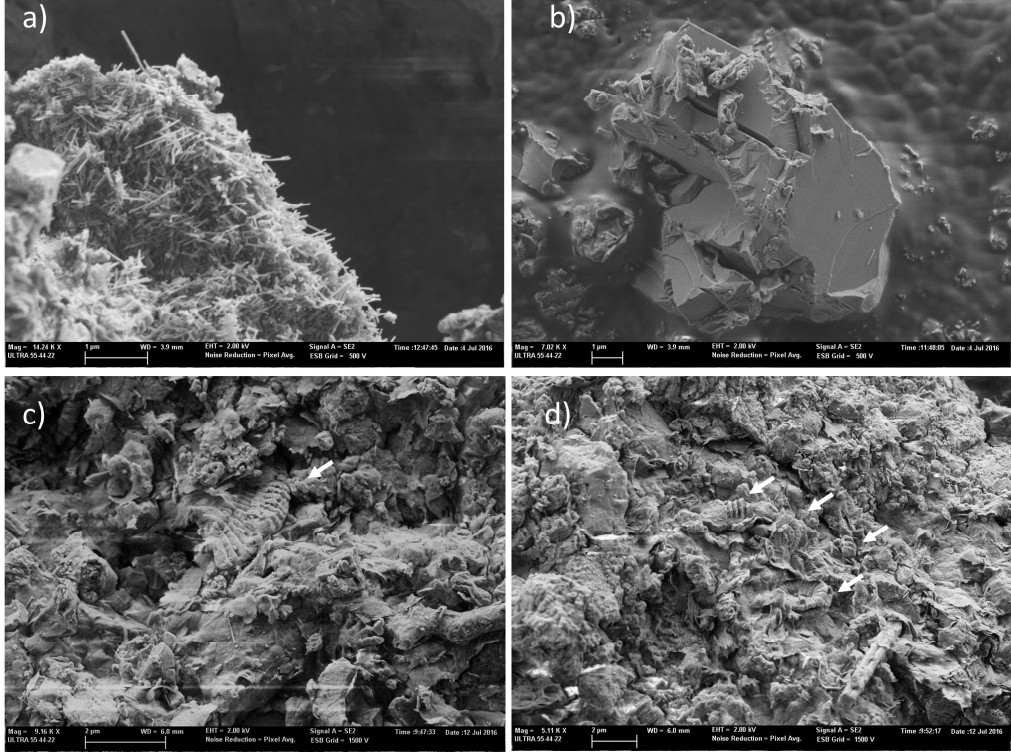

**Figure 7.** Secondary electron images acquired at 2 kV: (**a**) cluster of needles ascribed to halloysite clay found in sample; (**b**) siliceous mineral found in sample 6; (**c,d**) fragments of striated disks forming the cocosphere of cocolithophores in sample 0097. The arrows in the figures point out the striated disks.

Elemental compositions obtained are summarised in Table 1. The X-ray spectra acquired in small areas of the samples excised on the surface of the four sculptures provide an average value of the elemental chemical composition of the sculpture in the area where the remains of the pigment were placed (Table 1). The high content found in Ca that exhibits the X-ray spectra acquired in area mode is associated with calcite and dolomite from the biocalcarenite support. The occurrence of the emission lines of Mg, Al, Si, K, and Fe is ascribed to clayey minerals of the silicoaluminate type. The X-ray spectra acquired in Fe-rich grains found in each sample suggest that some cryptocrystalline haematite accompanies the clayey minerals that compose the pigment. These minerals play an important part in the dyeing properties of the red pigment [22]. P and S were only found in sample 0092. These elements are related to apatite and sulfate minerals present in the stone as accessory minerals or coming from the soil surrounding the sculptures in the burial. It is worth noting the appearance of the emission line of N in sample 0092, which is tentatively associated with organic compounds present on the surface of the sample as a result of a consolidation treatment carried out in the past (see FTIR results). This hypothesis is supported by the lower intensity of the X-ray signals observed in sample 0092 for the major components (Si, Ca and Fe), which may be associated with the barrier effect of X-ray absorption exerted by the protective layer of consolidant.

**Table 1.** Chemical composition calculated for the areas analysed: sample 6 (40 × 20 μm), sample 0092 (100 × 100 μm), sample 0093 (120 × 120 μm), and sample 0097 (40 × 20 μm). n.d.: not detected.

| Chemical Composition (wt%) | | | | | | | | |
|---|---|---|---|---|---|---|---|---|
| Sample | 6 | | 0092 | | 0093 | | 0097 | |
| Element | Area | Spot | Area | Spot | Area | Spot | Area | Spot |
| N | | | 7 (1) | | | | | |
| O | 38 (4) | 61.3 | 72 (5) | 71 (5) | 57 (5) | 65 (6) | 71 (5) | 54 (5) |
| Mg | 0.4 (0.1) | 0.9 (0.1) | 0.9 (0.1) | 1.3 (0.2) | 0.71 (0.09) | 1.3 (0.2) | 1.1 (0.1) | 0.73 (0.09) |
| Al | 1.1 (0.5) | 2.53 (0.03) | 1.04 (0.01) | 4.00 (0.04) | 1.88 (0.02) | 5.39 (0.05) | 4.06 (0.04) | 3.6 (0.1) |
| Si | 3.42 (0.03) | 6.86 (0.07) | 2.91 (0.03) | 11.4 (0.1) | 5.13 (0.05) | 21.5 (0.2) | 11.6 (0.1) | 13.8 (0.1) |
| P | n.d. | n.d. | 0.24 (0.05) | n.d. | n.d. | n.d. | n.d. | n.d. |
| S | n.d. | n.d. | 1.60 (0.05) | 0.77 (0.2) | n.d. | n.d. | n.d. | n.d. |
| K | 0.21 (0.06) | n.d. | n.d. | 0.46 (0.08) | 0.49 (0.01) | 1,4 (0.5) | 0.97 (0.03) | 1.67 (0.05) |
| Ca | 9.48 (0.09) | 7.99 (0.08) | 13.1 (0.1) | 8.37 (0.08) | 30.2 (0.3) | 3,78 (0.04) | 7.52 (0.08) | 3.25 (0.03) |
| Fe | 5.3 (0.4) | 20.4 (0.2) | 0.55 (0.03) | 2.30 (0.01) | 4.58 (0.01) | 2,18 (0.01) | 3.03 (0.01) | 22.2 (0.4) |

### 3.3. FTIR Spectroscopy

Figure 8 shows the IR absorption spectra obtained in the pigmented surface of two of the four samples studied (black line). The IR spectra of the stone material that makes up the bulk of the sculptures have also been included in Figure 8b–d (red line). The detail of the 500–900 cm$^{-1}$ region of the four IR spectra is shown in Figure 9.

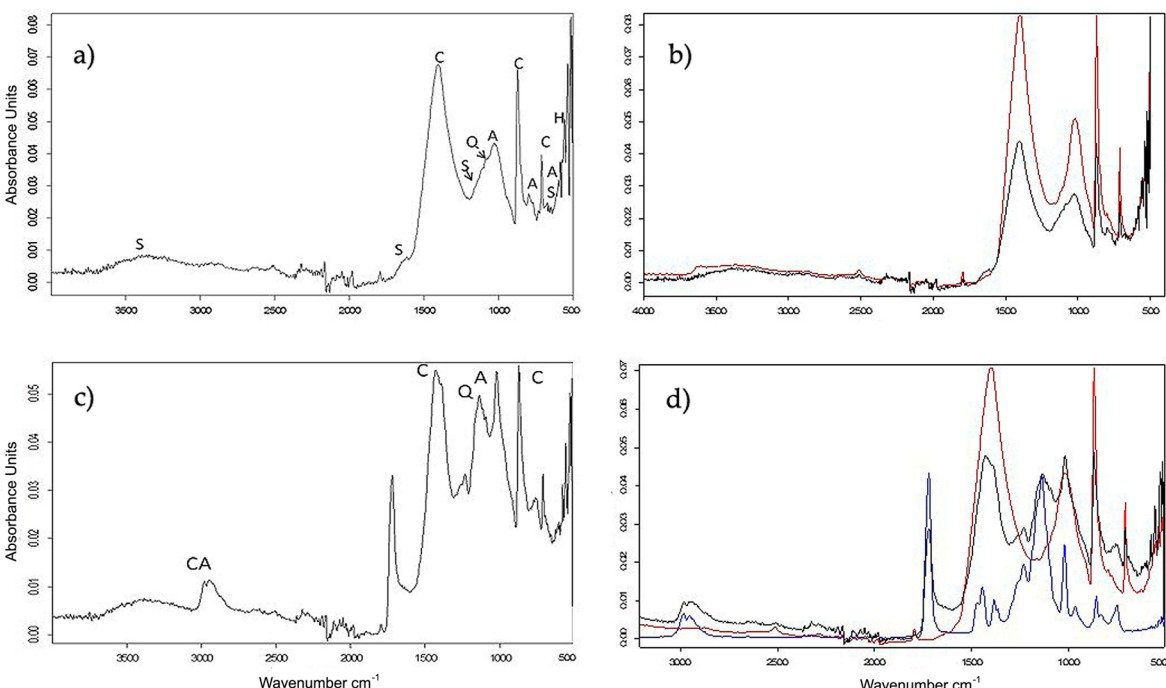

**Figure 8.** (**a**) IR spectrum of sample 0092; (**b**) IR spectra of sample 0092 (black line) and sample excised from the stone bulk of the sculpture (red line): A: clayey minerals, C: calcite, H: haematite, S: gypsum. (**c**) IR spectrum of sample 0097; (**d**) IR spectra of sample 0097 (black line), sample excised from the stone bulk of the sculpture (red line) and acrylic resin (blue line). A: clayey minerals, Q: quartz, C: calcite, CA: acrylic polymer.

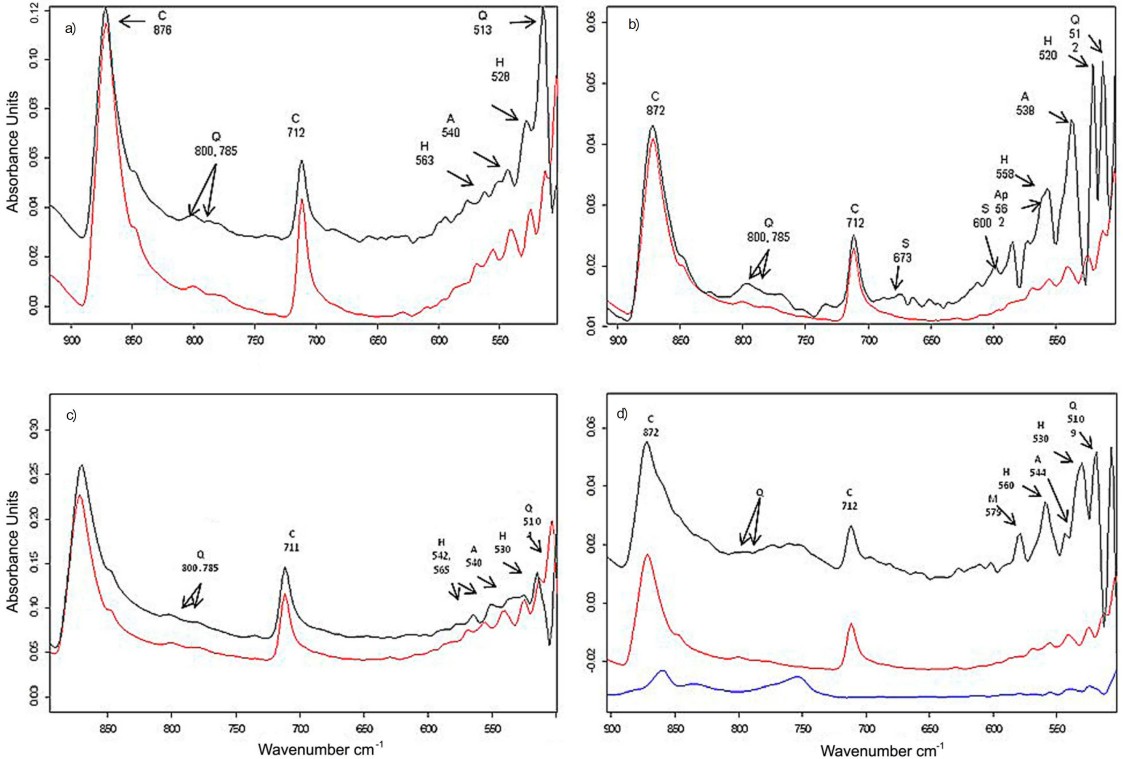

**Figure 9.** Detail of the 900–500 cm$^{-1}$ in the IR spectra: (**a**) sample 6; (**b**) sample 0092; (**c**) sample 0093 and (**d**) sample 0097 (black line); stone bulk (red line) and acrylic resin (blue line). A: clayey minerals, Q: quartz, C: calcite, H: haematite, S: gypsum; Ap: apatite mineral and M: magnetite.

The four IR spectra (Figures 8 and 9) are dominated by the IR bands of calcite (at ca. 1400, 872 and 710 cm$^{-1}$) and the bands of clayey minerals and quartz (at ca. 1000 cm$^{-1}$). The latter compounds also exhibit weak bands in the 3650–3700 cm$^{-1}$ region ascribed to stretching vibrations of the inner OH groups and sharp bands in the dioctahedral 1:1 clays (kaolinite group), which are common minerals found in many soils and rocks. These minerals are also recognised by sharp bands at around 799 and 777 cm$^{-1}$ [22]. Haematite is identified in the four sculptures via the IR band at ca. 560 and 530 cm$^{-1}$ associated with the stretching of the Fe-O bond in isodimensional and an-isodimensional particles, respectively [20]. Other compounds identified in specific samples were gypsum (bands at ca. 3400–3500, 1620, 1680, 1150, and 599, 670 cm$^{-1}$) and apatite (band at 560 cm$^{-1}$) in sample 0093. This latter band is in agreement with the 0.2% of P found in the elemental composition of this sample. Magnetite (band at 579 cm$^{-1}$) and acrylic resin (bands at 2983, 2947, 2872, 1722, 1422, 1387, 1235, 1143 and 1024 cm$^{-1}$), probably used as a consolidant, were identified in sample 0097. All the samples exhibit weak bands in the 2850–2980 cm$^{-1}$ range, ascribed to stretching vibrations of methyl and methylene groups in organic compounds. These bands are due to unspecific organic matter present on the surface of the buried sculptures and could also be ascribed to the remains of the consolidants presumably used in a prior conservation treatment. A summary of the IR bands of analytical interest identified in each sample and the corresponding compounds is given in Table 2 [25–31].

**Table 2.** IR bands of analytical interest identified in each sample and minerals identified.

| Minerals Identified | Sample | | | | IR Bands' Assignment (cm$^{-1}$) |
|---|---|---|---|---|---|
| | **6** | **0092** | **0093** | **0097** | |
| Calcite | X | X | X | X | 2873 ($2\nu_3$ mode of carbonate group) |
| | X | X | X | X | 2515 ($2\nu_2 + \nu_4$ mode of carbonate group) |
| | X | X | X | X | 1798 ($2\nu_1 + \nu_4$ mode of carbonate group) |
| | X | X | X | X | 1397-1408 ($\nu_3$ symmetric stretching of carbonate group) |
| | X | X | X | X | 870 ($\nu_2$ out of plane deformation of carbonate group) |
| | X | X | X | X | 711-2 ($\nu_4$ in-plane deformation of carbonate group) |
| Clayey minerals Quartz | X | | | | 3674 (stretching of inner-surface hydroxyl groups located at the surface of the octahedral sheet of the layers in dioctahedral 1:1 clays) |
| | X | | X | | 3621-26 (stretching of inner hydroxyl groups bonded to octahedral cations in dioctahedral 1:1 clays) |
| | X | X | X | X | 3397-3400 (stretching of surface hydroxyl groups of adsorbed water) |
| | X | X | X | X | 1084 (shoulder) (in phase stretching band of apical Si-O bonds) |
| | X | X | X | X | 1020-27 (antisymmetric in-plane stretching of the Si-O-Si group) |
| | X | X | X | X | 910 (deformation of inner and inner-surface OH groups) |
| | X | X | X | X | 795-800, 775-784 (perpendicular Si-O deformations of the surface hydroxyl layer) |
| Haematite | X | X | X | X | 520-65 (isodimensional and an-isodimensional stretching of Fe-O bond) |
| Magnetite | | | | X | 579 (stretching of Fe-O bond) |
| Acrylic resin | X | X | X | X | 2983, 2947, 2872 (stretching of CH$_2$ and CH$_3$ group) |
| | | | | X | 1722 (stretching of C=O ester group) |
| | | | | X | 1422 (shoulder) (bending -CH$_2$ group) |
| | | | | X | 1387 (shoulder) (bending -CH$_3$ group) |
| | | | | X | 1235 (stretching C-O-C group) |
| | | | | X | 1143 (stretching C-O-C group) |
| | | | | X | 1024 (stretching CH$_3$-COO- group) |
| Gypsum | | X | | | 3500-3400 (stretching OH group) |
| | | X | | | 1680, 1620 (deformation OH group) |
| | | X | | X | 1115 ($\nu_3$ stretching of sulfate group) |
| | | X | | | 673, 600 ($\nu_2$ stretching of sulfate group) |
| Apatite minerals | | X | | | 562 $\nu_4$ stretching of apatite group |

## 4. Discussion

In the sculptures of Cerrillo Blanco, red crystalline aggregates have been deposited between the characteristic pores of the calcarenite, demonstrating the presence of reddish polychromy. The analyses indicate that they are clayey materials with a high iron content.

In the SOM/PM microscopy study, the presence of a preparation layer has not been observed, unlike what has been described in other Iberian sculptures such as the *Lady of Baza* [32]. Nevertheless, in sample 0092, the FTIR analysis indicates the presence of gypsum, which could be related to a preparatory coating prior to the polychromy. The long time spent underground could have caused the stucco layer to disappear due to material migration and recrystallisation processes related to changes in environmental humidity. A similar phenomenon has been described in the *Lady of Elche*. Luxán et al. [33] maintain that the gypsum particles found in this sculpture's paint layer are the result of a recrystallisation process through which the gypsum went from the stone or from the stucco to the polychrome.

Nor have traces of organic matter been found that might indicate the presence of a binder, as has occurred in cave paintings and other artistic manifestations from Antiquity. This could be explained by its disappearance during the time it remained buried or simply because the pigment was applied directly onto the substrate or in an aqueous or aqueous dispersion, indistinguishable from the calcium carbonate in the support using the analytical methods. In this case, the environmental conditions during the time buried could have fostered precipitation in the form of calcite crystals, which would fix the pigment particles, acting as an inorganic binder.

One could also consider the possible masking of the binder residue by using acrylic resin as a restoration treatment. Lastly, there is another possible factor to take into account: the process of intentional destruction suffered by the sculptures, which could have contributed to the loss of the original polychromy. However, in the fine polishing finish seen on the pieces, there are no marks of scraping or chipping on the surface that might indicate stripping of the paint.

As for the symbolism and purpose of polychromy in Iberian sculptures, little has been known until now. In this case, only the remains of red polychrome have been found, unlike other Iberian sculptures such as the *Ladies* of Elche and Baza [14,32,33], on which other pigments such as Egyptian blue or cinnabar have been described, which have not been documented in this case.

The reddish monochrome paint described here is not an isolated case in Iberian art. Similar pictorial techniques have been documented in other sculptural ensembles, such as the *Dama oferente* (*Offering Lady*) at Cerro de los Santos, which shows traces of red polychrome applied directly onto the support; the sphinxes in semi-relief from El Salobral (Albacete); the anthropomorphic head-stele from Villaricos; and the warriors on the corner ashlar from Osuna [13,34]. In other carvings, a stucco base prior to polychrome has been documented, such as in the *Guerrero de Baza* (*Warrior of Baza*). In this case, the painting contains red earth, cinnabar and traces of ash from animal bones as a colouring matter to achieve various reddish and mauve tones [35].

The presence and permanence of iron oxides can be explained by the fact that they rae very stable, abundant, and easily usable pigments, whose use has been documented since the Palaeolithic era. What seems clear is that the schematic use of colours seen in Iberian sculptures does not seem to seek any intention of realism but rather responds to a symbolic motivation. In the case of the sculptural ensembles from Cerrillo Blanco, the red monochromatic polychromy was able to enhance the characteristics and attributes of the warriors and zoomorphs as an exaltation and homage to the military power of the aristocracy of the time, whereas the female sculptures such as the *Ladies of* Baza and Elche used a broader colour palette, polychroming their jewellery and clothing to show their social position and wealth.

## 5. Conclusions

1.  The study carried out shows that many of the Cerrillo Blanco ensemble of sculptures had polychromy on them. The pigment used was a red clay. No remains of other pigments have been found, nor of organic compounds typical in materials used as binders in Prehistory.
2.  No preparation layer prior to the polychrome has been observed under the microscope. However, the FTIR analysis indicates the presence of gypsum in one of the samples, so the initial existence of a stucco coating, which was lost over the time the sculptures remained buried, cannot be ruled out.
3.  An acrylic resin has been identified in two samples as a result of restoration tasks that do not seem to have been justified, because the sculptures are in a good state of conservation and are kept in controlled interior spaces.
4.  The reddish monochrome paint described on the Cerrillo Blanco sculptures coincides with the kind described in other representative sculptural groups of the Iberian period, though very little is known about its meaning and symbolism.

**Supplementary Materials:** The supporting information can be downloaded at https://www.mdpi.com/article/10.3390/coatings13101798/s1. Table S1: Average relative standard deviation values (RSD) calculated for the samples analyzed; Table S2: Relative deviation (RD) calculated for the reference material SRM 679-Brick clay in the experimental conditions used in this study. (Refs [36–38] are included in the Supplementary Materials file).

**Author Contributions:** Conceptualisation and work design, M.B.R.-R., M.T.D.-C. and J.R.-N.; methodology and analyses, M.T.D.-C. and M.B.R.-R.; writing—original draft preparation review and editing, J.R.-N.; project administration, funding acquisition, F.B.-G. All authors have read and agreed to the published version of the manuscript.

**Funding:** This research was funded by the following projects: ECODIGICOLOR, grant number TED2021-132023B-I00, supported by MCIN/AEI /10.13039/501100011033 and Unión Europea NextGenerationEU/ PRTR (Proyectos estratégicos orientados a la transición ecológica y digital), project BIOALHAMBRA, grant number PID2022-143064OB-I00, supported by MCIN/ AEI/10.13039/ 501100011033 «Proyectos de Generación de Conocimiento», project FICOARTE 2, grant number P18-FR-4477, supported by "Consejería de Universidad, Investigacion e Innovación", Junta de Andalucía, Programa FEDER, "Andalucía se mueve con Europa", and Grant PID2020-113022GB-I00 funded by MCIN/AEI.10.13039/501100011033 and by "ERDF A way of making Europe", by the "European Union".

**Institutional Review Board Statement:** Not applicable.

**Data Availability Statement:** Data are contained within the article or Supplementary Material.

**Acknowledgments:** We would like to thank the Museo Provincial de Jaén for their logistical support, without which it would not have been possible to do this work. We would also like to thank Luis Emilio Vallejo Delgado, Director of the Obulco Municipal Archaeological Museum (Porcuna, Spain), for his support in this research.

**Conflicts of Interest:** The authors declare no conflict of interest.

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
