# Peer review of "Polychromy in the Iberian Sculptures of Cerrillo Blanco: Analytical Study, Historical Context and State of Conservation"

_coatings, doi:10.3390/coatings13101798_

Round 1

Reviewer 1 Report

Dear Editor,

I would like to express my sincere gratitude for the opportunity to review the manuscript entitled " Polychromy in the Iberian sculptures of Cerrillo Blanco. Analytical study, historical context and state of conservation".

The subject matter addressed in the manuscript is highly significant and of great interest to the field. However, it is worth noting that the authors have based their conclusions on a limited number of samples, which may impact the generalizability of their findings.

I would like to commend the authors for the well-organized and concise introduction, which includes several pertinent bibliographical citations. The "Materials and Methods" section is similarly well-structured and presents the information in a clear and understandable manner.

Regarding the "Results" section, I believe that the text related to the microscopic description and SEM-EDS results is somewhat sparse, and the quality of the images presented is suboptimal. I would therefore like to request that the authors include images with improved resolution in the text and provide indications within the images of the different parts described in the text, if possible.

Furthermore, I recommend that the caption of Table 1 be rephrased to enhance clarity for the reader. Additionally, the text from lines 166 to 177 requires greater clarification and additional bibliographical references to support the authors' assertions.

I would also like to commend the authors on their presentation of the FTIR analysis results, which I found to be well-executed and informative.

In the text, the authors refer to the presence of acrylic resins in two of the four specimens analyzed. I wonder if it is possible to provide further indications, such as whether the specimens where acrylic resin was found display a different appearance from those that do not, or whether any restoration interventions involving the use of acrylic resin are documented.

Author Response

Reviewer 1

Dear Editor,

I would like to express my sincere gratitude for the opportunity to review the manuscript entitled " Polychromy in the Iberian sculptures of Cerrillo Blanco. Analytical study, historical context and state of conservation". 

The subject matter addressed in the manuscript is highly significant and of great interest to the field. However, it is worth noting that the authors have based their conclusions on a limited number of samples, which may impact the generalizability of their findings. 

Answer: What you mention in your comment is very accurate. The limitation in sampling is a common problem when carrying out research on real heritage, as in this case, and which is also of special historical and artistic relevance. We have taken the samples that the Jaén museum allowed us to take. We would have liked to do a broader sampling but it was not possible. Anyway, we carried out a thorough inspection of the pieces displayed in the room and the fragments preserved in the deposit in search of traces of polychromy, and in no case we found traces of a color other than the reddish one analyzed, so we believe that the analyzed samples are representative of the whole.

I would like to commend the authors for the well-organized and concise introduction, which includes several pertinent bibliographical citations. The "Materials and Methods" section is similarly well-structured and presents the information in a clear and understandable manner.

Answer: Thak you very much for your comments

Regarding the "Results" section, I believe that the text related to the microscopic description and SEM-EDS results is somewhat sparse, and the quality of the images presented is suboptimal. I would therefore like to request that the authors include images with improved resolution in the text and provide indications within the images of the different parts described in the text, if possible. Furthermore, I recommend that the caption of Table 1 be rephrased to enhance clarity for the reader. Additionally, the text from lines 166 to 177 requires greater clarification and additional bibliographical references to support the authors' assertions.

Answer: According to the reviewer’s comment the discussion of the OM and SEM-EDX results has been enlarged. In addition, new electron images with better quality have been provided and a more complete description of the working conditions of the SEM-EDX has been provided as supplementary electronic material. Additionally, a paragraph with a brief description of the procedure used for taking the pictures has been included in the revised version of the manuscript and more additional details on the working conditions used are summarized in table 1S provided as supplementary electronic material. Figure 2 has been modified to improve its quality too. Table 1 caption has been modified for clarity: Table 1. Chemical composition calculated for the areas analysed: sample 6 (40x20 m), sample 0092 (100x100 m), sample 0093 (120x120 m) and sample 0097 (40x20 m).

I would also like to commend the authors on their presentation of the FTIR analysis results, which I found to be well-executed and informative.

Answer: Thak you very much for your comments

In the text, the authors refer to the presence of acrylic resins in two of the four specimens analyzed. I wonder if it is possible to provide further indications, such as whether the specimens where acrylic resin was found display a different appearance from those that do not, or whether any restoration interventions involving the use of acrylic resin are documented.

Answer: The use of acrylic resins in previous interventions has been documented in Espinosa-Gaitán et al (2001), cited in the bibliography. Likewise, in a recent work published by our research group (Romero-Noguera et al 2023) dedicated to the composition of the material, data in this regard is attached. A comment and bibliographical references have been added.

Reviewer 2 Report

The authors have investigated Polychromy in Iberian sculpture using petrographic analysis, SEM/EDX and FTIR spectroscopy. Insights of unique samples are obtained, and the manuscript is generally well-written. There are only 6 issues that the authors should consider before publication to further improve the paper.

1.       It would be beneficial for the readers to have explanations whenever different conditions are used to characterize each sample. For example, c) 0093 using polarized light with Nicols partially crossed (75 degree), and d) sample 0097 using polarised light with Nicols partially crossed (80 degree).

2.       Why is the EDX carried out on different sizes of each sample? Do author measure different points on each sample and find the average composition?    

3.       It seems that the presence of biological structures in sample 0097 is insufficiently supported by the results provided. The authors have identified them by pointing to a feature which is barely seen in the micrograph. More supporting results or careful interpretations are needed.

4.       The comparison to other sculptures makes this manuscript interesting. Is it possible to add other works to “In the SOM/PM microscopy study, the presence of a preparation layer has not been observed, unlike what has been described in other Iberian sculptures such as the Lady of Baza [30].” and “Similar pictorial techniques have been documented in other sculptural ensembles, such as the Offering Lady at Cerro de los Santos, which shows traces of red polychrome applied directly onto the support; the sphinxes in semi-relief from El Salobral (Albacete), the anthropomorphic head-stele from Villaricos and the warriors on the corner ashlar from Osuna [13, 32].?

5.       Full stop (.) should be replaced by Colon (:) in the title.

6.       The authors should carefully proofread the manuscript to eliminate typographic errors and ensure consistency. Here are a few examples:

•        For 2×10–9A in Page 4, “–9” should be in superscript.

•        In [26], (2003) after the author list should be removed.

Addressing these minor revisions will enhance the overall quality of the paper.

The manuscript is generally well-written. 

Author Response

Reviewer 2

The authors have investigated Polychromy in Iberian sculpture using petrographic analysis, SEM/EDX and FTIR spectroscopy. Insights of unique samples are obtained, and the manuscript is generally well-written. There are only 6 issues that the authors should consider before publication to further improve the paper.

  1. It would be beneficial for the readers to have explanations whenever different conditions are used to characterize each sample. For example, c) 0093 using polarized light with Nicols partially crossed (75 degree), and d) sample 0097 using polarised light with Nicols partially crossed (80 degree).

Anwer: For improving the quality of the image, the stage of the microscope was rotated until maximum bright of the red grains of pigment. Spinning angle was measured using as reference a quartz grain of the mounted sample placed in the centre of the stage at extinction angle. According to the reviewer’s comment, a paragraph with the description of the procedure used for taking the pictures has been included in the revised version of the manuscript.

  1. Why is the EDX carried out on different sizes of each sample? Do author measure different points on each sample and find the average composition?    

Answer: In the heterogeneous samples composed of several minerals, X-ray spectra were acquired in area mode. The acquisition area varied in order to maintain always it over the pattern of heterogeneities of the stony material. This procedure is accompanied of spot analysis in individual grains of the minerals composing the outer and irregular pigment layer. According to the reviewer’s comment.

  1. It seems that the presence of biological structures in sample 0097 is insufficiently supported by the results provided. The authors have identified them by pointing to a feature which is barely seen in the micrograph. More supporting results or careful interpretations are needed.

Answer: According to the reviewer’s comment, new electron images and references have been provided that properly sustain the identification of microfossil remains in the stony support of the sculpture.

  1. The comparison to other sculptures makes this manuscript interesting. Is it possible to add other works to “In the SOM/PM microscopy study, the presence of a preparation layer has not been observed, unlike what has been described in other Iberian sculptures such as the Lady of Baza [30].” and “Similar pictorial techniques have been documented in other sculptural ensembles, such as the Offering Lady at Cerro de los Santos, which shows traces of red polychrome applied directly onto the support; the sphinxes in semi-relief from El Salobral (Albacete), the anthropomorphic head-stele from Villaricos and the warriors on the corner ashlar from Osuna [13, 32].?

Answer: There aren't many more examples to comment on. However, we have added another important one, such as the Warrior of Baza. In this case the painting contains red earth, cinnabar and traces of ash from animal bones as a colouring matter to achieve various reddish and mauve tones. A new reference has been added.  

  1. Full stop (.) should be replaced by Colon (:) in the title. DONE
  2. The authors should carefully proofread the manuscript to eliminate typographic errors and ensure consistency. Here are a few examples:
  • For 2×10–9A in Page 4, “–9” should be in superscript. DONE
  • In [26], (2003) after the author list should be removed. DONE.

The manuscript has been throughly revised in order to eliminate this errors. Thank you very much for your comments.

Addressing these minor revisions will enhance the overall quality of the paper.